# Effect of mediolateral leg perturbations on walking balance in people with chronic stroke: A randomized controlled trial

**Alexa A. Krause**[1], **Nicholas K. Reimold**[1], **Aaron E. Embry**[1,2], **Heather L. Knight**[1], **Camden J. Jacobs**[1], **Andrea D. Boan**[1], **Jesse C. Dean**[1,2]*

**1** Medical University of South Carolina, Charleston, SC, United States of America, **2** Ralph H. Johnson VA Health Care System, Charleston, SC, United States of America

* deaje@musc.edu

## Abstract

Many people with chronic stroke (PwCS) exhibit deficits in step width modulation, an important strategy for walking balance. A single exposure to swing leg perturbations can temporarily strengthen this modulation. The objective of this parallel, double-blinded, randomized controlled trial was to investigate whether repeated perturbations cause sustained increases in step modulation (NCT02964039; funded by the VA). 54 PwCS at the Medical University of South Carolina were randomly assigned to one of three intervention groups: Control (n = 18), with minimal forces; Assistive (n = 18), pushing the swing leg toward a mechanically appropriate location; Perturbing (n = 18), pushing the swing leg away from a mechanically appropriate location. All intervention groups included 24 training sessions over 12-weeks with up to 30-minutes of treadmill walking while interfaced with a novel force-field and a 12-week follow-up period, with five interspersed assessment sessions. Our primary outcome measure was paretic step width modulation, the partial correlation between step width and pelvis displacement ($\rho_{SW}$). Secondarily, we quantified swing and stance leg contributions to step modulation, clinical assessments of walking balance and confidence, and real-world falls. Outcomes were analyzed for participants who completed all assessment sessions (n = 44). Only the Perturbing group exhibited significant increases in paretic $\rho_{SW}$, which were present after 4-weeks of training and sustained through follow-up (t = 2.42–3.17). These changes were due to improved control of paretic swing leg positioning. However, perturbation-induced changes in step modulation were not always significantly greater than those in the Control group, and clinical assessments were similar across intervention groups. Participants in the Perturbing group experienced a lower fall rate than those in the Control group (incidence rate ratio = 0.53), although our small sample size warrants caution. The present results indicate that perturbations can cause sustained modifications of targeted biomechanical characteristics of post-stroke gait, although such changes alone may be insufficient to change more complex clinical assessments.

**Data Availability Statement:** All relevant data are within the manuscript and its Supporting Information files.

**Funding:** This work was supported in part by Merit Award Number I01 RX002256 (JCD) from the United States (U.S.) Department of Veterans Affairs, Rehabilitation Research and Development (https://www.research.va.gov/). This work was also supported in part by the South Carolina Research Center for Recovery from Stroke COBRE through grant P20 GM109040 from the National Institutes of Health (https://www.nih.gov/). The funders did not play any role in study design, data collection and analysis, decision to publish, or preparation of the manuscript.

**Competing interests:** The authors have declared that no competing interests exist.

## Introduction

People with chronic stroke (PwCS) often experience walking balance deficits, here defined as a reduced ability to walk in their environment without falling. The rate of falls in community ambulating PwCS is more than double that of age-matched controls [1], and walking is the most common activity at the time of a fall [1]. Additionally, many PwCS exhibit a fear of falling and reduced balance self-efficacy, which has been linked to reduced functional mobility [2].

Appropriately modulating step location is one important strategy used to ensure mediolateral balance [3]. In neurologically-intact individuals, step width scales with pelvis motion [4,5]; larger mediolateral pelvis displacements or velocities away from the stance foot tend to be accompanied by wider steps. This relationship can be quantified using the partial correlation between step width and mediolateral pelvis displacement at the start of the step ($\rho_{SW}$), accounting for pelvis velocity [6]. In PwCS, this metric of step width modulation is significantly lower for paretic steps than non-paretic steps [7], suggesting an altered strategy underlying the control of step location for the more limited leg.

A reduced ability to appropriately modulate mediolateral step location may be linked to poor walking balance in PwCS. We recently found that $\rho_{SW}$ for paretic steps was significantly (albeit moderately) correlated with Functional Gait Assessment (FGA) score [8], a common clinical assessment of walking balance. A similar metric of paretic foot placement modulation relative to the pelvis (focusing on swing leg control) was also significantly related to Activities-specific Balance Confidence (ABC) score (a clinical measure of balance self-efficacy), fear of falling, and self-reported fall history [8].

The purpose of this study was to investigate whether the modulation of mediolateral paretic step location can be strengthened through repeated exposure to a novel mechanical environment, thus improving post-stroke walking balance. We previously found that a force-field designed to push the swing leg toward targeted mediolateral locations [9] can effectively influence mediolateral step modulation in neurologically-intact individuals [10]. This influence can take the form of either *assisting* appropriate step locations (increasing $\rho_{SW}$) or *perturbing* appropriate step locations (decreasing $\rho_{SW}$). The perturbation approach elicited after-effects in which $\rho_{SW}$ increased relative to baseline once the perturbations ceased, providing evidence of sensorimotor adaptation [10]. These after-effects were due to adjustments in foot placement modulation relative to the pelvis [11] and have been observed for paretic steps in PwCS following a single exposure to the same perturbation paradigm [12]. However, it is presently unclear whether repeated exposure to such perturbations can elicit motor learning [13], whereby step modulation remains strengthened over extended time periods. Supporting the potential of this general approach, repeated exposure to the novel mechanical environment of split belt walking [14] has been followed by sustained improvements in step length asymmetry in PwCS [15,16].

Our primary hypothesis was that repeated exposure to appropriately targeted perturbations would cause sustained increases in paretic step width modulation ($\rho_{SW}$) among PwCS. In contrast, we did not expect that repeated exposure to either assistance or minimal external forces (a control group) would evoke such changes in paretic $\rho_{SW}$. Exploratory analyses compared changes in paretic $\rho_{SW}$ between the control group and both the perturbing and assistive groups. Secondarily, we hypothesized that repeated perturbations would cause sustained increases in paretic foot placement modulation relative to the pelvis, as well as improvements in clinical assessments related to walking balance and confidence (FGA score, ABC score, and fear of falling). As with our primary metric, we performed exploratory comparisons of these outcomes (as well as real-world falls during the 12-week intervention and a 12-week follow-up period) between the control group and both the perturbing and assistive groups.

## Materials and methods

### Trial design

This study used a parallel arm design, with participants randomly assigned to one of three study arms. The three arms were Control, Assistive, and Perturbing, as detailed below. The allocation ratio was 1:1:1 across these three arms. Our description of this study follows the Consolidated Standards of Reporting Trials (CONSORT) guidelines. This study was preregistered on clinicaltrials.gov (NCT02964039). Participant flow through the study is illustrated in Fig 1 and described in more detail in the recruitment and participant flow section below.

### Participants

All participants provided written informed consent using a document approved by the Medical University of South Carolina (MUSC) Institutional Review Board, and consistent with the Declaration of Helsinki. Participants were recruited from a database housed at MUSC. which contains contact information and basic health and demographic information for individuals who have experienced a stroke and have agreed to be contacted for research participation. Whether potential participants met the study inclusion and exclusion criteria was determined

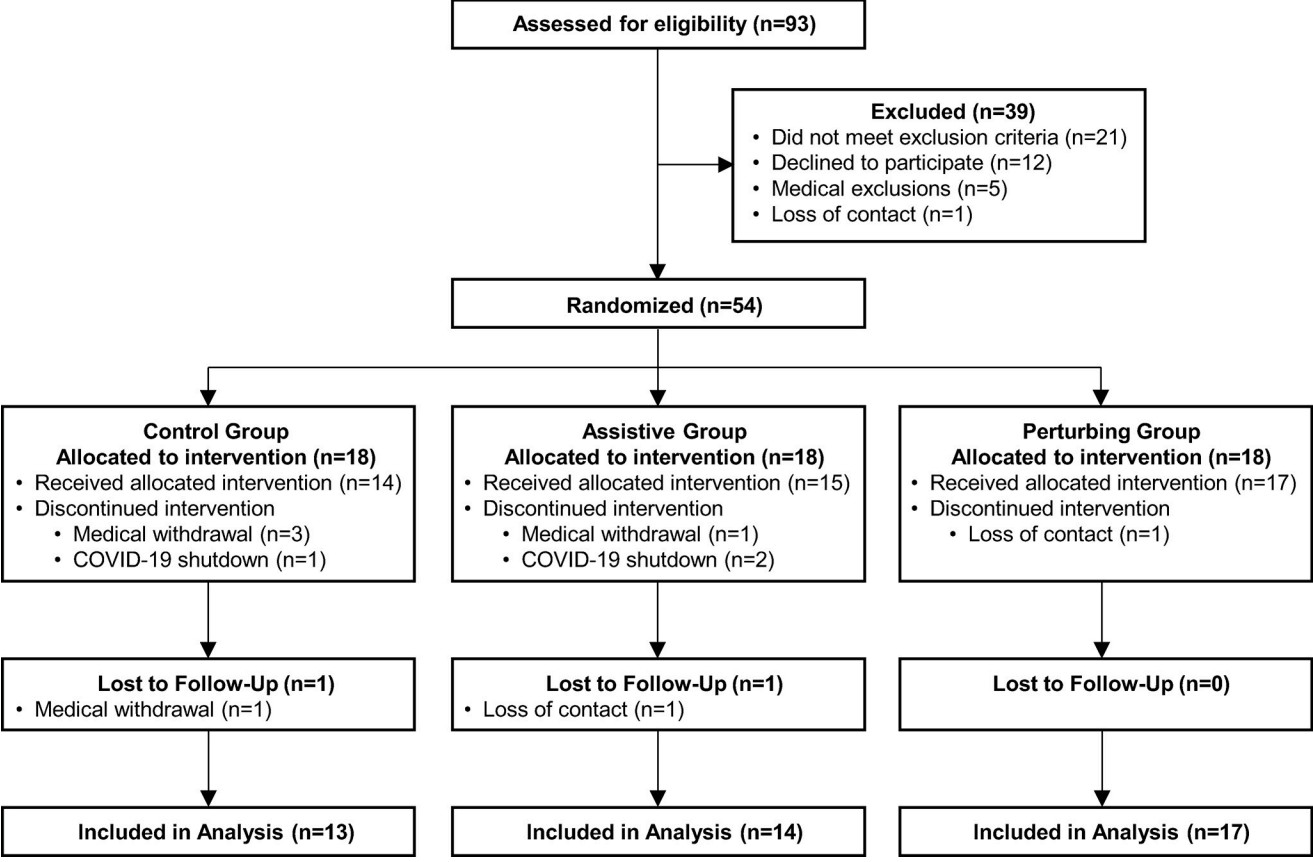

**Fig 1. Participant flow through the study.** Prior to randomization, five individuals were excluded for medical reasons that were not part of our pre-planned exclusion criteria, but would have affected their ability to participate in the intervention: Upcoming arm surgery, upcoming toe amputation, upcoming Achilles tendon surgery, anticipated Botox treatments, and undisclosed upcoming medical treatment. Three participants were medically withdrawn from the Control group during intervention for the following reasons: Wrist injury from a car accident, experience of a stroke, hospitalization for pneumonia. One participant was medically withdrawn from the Assistive group during intervention due to being diagnosed with Parkinson's disease. One participant was medically withdrawn from the Control group during the follow-up period due to experiencing a stroke.

during an initial screening session. The study inclusion and exclusion criteria were chosen to include a broad range of post-stroke individuals likely to receive physical therapy focused on walking balance (e.g., having some walking ability), while simultaneously reducing the risk of negative health events that could be caused by participation in a study that involves periods of exercise.

Inclusion criteria were: 1) age $\geq$ 21 years; 2) experience of a stroke at least 6 months prior to participation; 3) overground gait speed of at least 0.2 m/s; 4) ability to walk at self-selected speed for at least 3-minutes without a cane or walker; 5) paretic $\rho_{SW}$ value (described in the Introduction and detailed below) of below 0.56, less than the 5th percentile value for this metric among neurologically-intact individuals walking at a typical preferred speed [6]; 6) provision of informed consent.

Exclusion criteria were: 1) resting heart rate above 110 beats/min; 2) resting blood pressure higher than 200/110 mm Hg; 3) history of congestive heart failure, unstable cardiac arrhythmias, hypertrophic cardiomyopathy, severe aortic stenosis, angina, or dyspnea at rest or during activities of daily living; 4) pre-existing neurological disorders or dementia; 5) history of major head trauma; 6) legal blindness or severe visual impairment; 7) history of deep vein thrombosis or pulmonary embolism within 6 months; 8) uncontrolled diabetes with recent weight loss, diabetic coma, or frequent insulin reactions; 9) orthopedic injuries or conditions with the potential to alter the ability to adjust foot placement while walking. An originally planned exclusion criterion of life expectancy less than one year was removed from the study before enrollment began.

Participants received remuneration of $50 for each study session they attended. All study procedures were conducted in Locomotion Energetics and Assessment Laboratory of the MUSC College of Health Professions Research Building, in Charleston, South Carolina.

## Interventions

All three intervention groups followed the same overall structure, schematically illustrated in Fig 2A. The interventions consisted of 24 training sessions, scheduled twice a week for 12 weeks. Assessment sessions were interspersed at five time points in the intervention: Pre-Intervention (within one week prior to the first training session); 4-Week (after 4 weeks of training); 8-Week (after 8 weeks of training); Post-Intervention (within one week after the final training session); Follow-up (12-weeks following the Post-Intervention session).

Training sessions were identically structured across all three intervention groups, as illustrated in Fig 2B. Participants first performed a 3-minute overground walking trial, in which they walked at their self-selected speed back and forth along a 10-meter path. Participants then performed a 3-minute treadmill walking trial at their self-selected speed in which they were not interfaced with the force-field. The self-selected speed was identified in each training session by gradually increasing the treadmill speed in increments of 0.05 m/s until participants reported that the speed was faster than they would normally walk around their house or a store. All subsequent trials used the speed just below the final level. Participants performed up to ten 3-minute trials in which they walked at their self-selected speed while interfaced with our force-field. The 3-minute trial duration was chosen based on preliminary work with similar inclusion criteria, in which participants were able to perform trials of this duration without requiring rest breaks. The total training time of 30 minutes was based on prior studies in which this training duration was sufficient to elicit adaptive gait changes during split-belt walking [15], walking with added leg mass [17], and a novel gait balance task [18]. Treadmill trials were terminated if participants reported being too fatigued to continue, were unable to keep up with the treadmill speed, or if blood pressure during rest periods exceeded 200/110

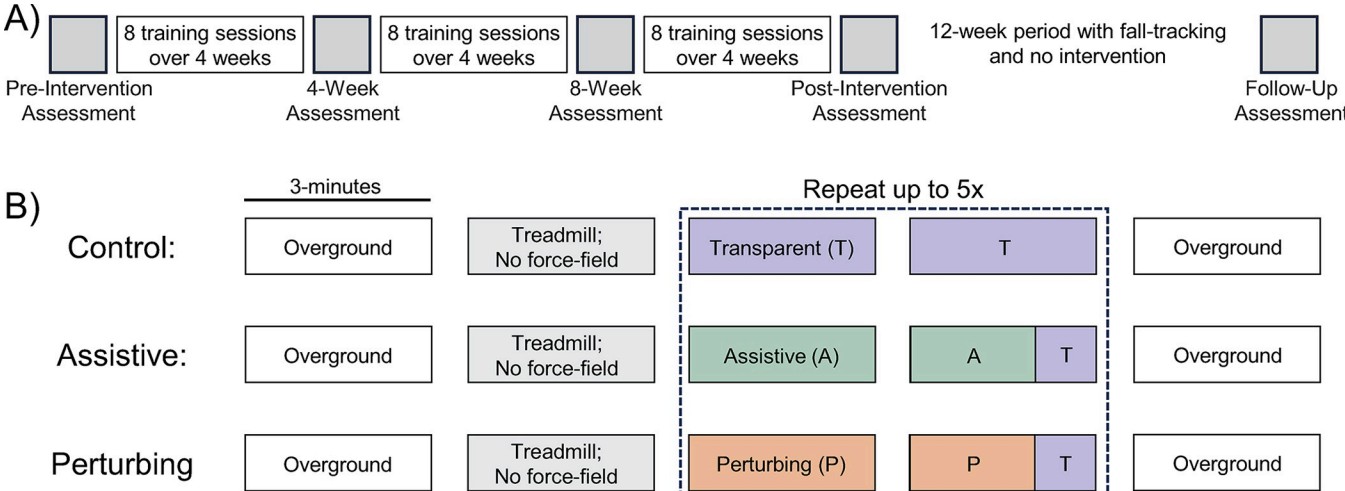

**Fig 2. Schematic diagrams of the intervention structure.** A) For all three intervention groups, participants completed five assessment sessions interspersed throughout a 12-week period of 24 training sessions and a 12-week follow-up period in which we tracked falls. B) The structure of the training sessions was the same for all three intervention groups. Participants performed up to thirteen 3-minute walking trials and were interfaced with the force-field for up to ten of these trials. The groups differed only in terms of the force-field control mode (Transparent [T], Assistive [A], or Perturbing [P]) during these trials. For both the Assistive and Perturbing groups, the force-field was in Transparent mode for the final minute of even numbered trials.

mm Hg. Therefore, the maximum total training time of 30 minutes was not achieved in all sessions. A final 3-minute overground walking trial at self-selected speed was performed at the end of the session, an approach suggested to improve real-world retention of gait changes [13]. Rest breaks were provided as needed between trials. In all overground and treadmill walking trials, participants wore a harness attached to an overhead rail. The harness did not support body weight but would prevent participants from falling to the ground in case of a loss of balance. Participants were not permitted to hold onto a handrail during walking trials. For all trials, one experimenter operated the computer that controlled the treadmill and data collection software, while another experimenter was next to the participant in case of a loss of balance.

Participants in all three intervention groups interfaced with a force-field able to exert mediolateral forces on the legs while walking. This device has been described in detail elsewhere [9]. Briefly, participants wore leg cuffs on the lateral side of their shanks. These leg cuffs interfaced with wires running parallel to the treadmill belts and allowed free anteroposterior and vertical leg motion. The end points of these wires could be repositioned in near real-time as participants walked to exert medial or lateral forces on the swing leg. The resultant mediolateral leg forces were proportional to the leg's mediolateral deviation from the targeted position, with an effective stiffness of 180 N/m. This stiffness level was previously used to successfully influence foot placement in both neurologically-intact controls [10,11] and PwCS [12].

This study involved three force-field control modes described previously [10]: Transparent (previously also termed Control), Assistive (previously also termed Error Reduction), and Perturbing (previously also termed Error Augmentation). In the Transparent mode, the force-field wire attachment points remained mediolaterally aligned with the corresponding leg cuff worn by participants, following near real-time measurements of the mediolateral position of LED markers placed on the leg cuffs. The Transparent mode thus minimized the mediolateral forces acting on the legs to have a minimal impact on stepping behavior [10]. In the Assistive mode, at the start of each step, our data collection system predicted a mechanically appropriate step width based on the mediolateral pelvis displacement from the stance foot, following speed-dependent relationships observed in neurologically-intact controls without balance

deficits [6,12]. These predictions accounted for each individual participant's baseline average step width, such that participants would not be encouraged to walk with either overly narrow or overly wide steps outside of their normal step width range [12]. The force-field then pushed the swing leg mediolaterally toward the position that would achieve this step width. Therefore, users experienced forces that *assisted* their performance of the targeted gait behavior, albeit potentially in the form of foot placement locations that differed from their baseline gait pattern. In the Perturbing mode, we again predicted a mechanically appropriate step width at the start of each step, but used the force-field to push the swing leg mediolaterally away from this step width. For example, if a wide step width was predicted, the swing leg was pushed in a medial direction, encouraging a narrower step width.

In the Control group, the force-field remained in Transparent mode during all trials in which participants interfaced with the force-field. In the Assistive group, the force-field was in Assistive mode for most of the walking time. In alternating trials (starting with the second force-field trial), the force-field changed to Transparent mode for the last minute of the 3-minute walking period (see Fig 2B). These "catch periods" were included for two reasons. First, their inclusion will allow future exploratory analyses to investigate the time course of potential changes in step width modulation during training sessions. Second, retention of a learned movement pattern may be enhanced when exposure to a novel mechanical environment includes periods in which this novel environment is removed, as recently observed for PwCS experiencing pelvis perturbations while walking [19]. In the Perturbing group, the force-field was in Perturbing mode other than the last minute of alternating trials, when it was in Transparent mode.

The five Assessment sessions followed a consistent structure. A trained, licensed physical therapist or physical therapist assistant performed a series of clinical assessments, as detailed in the Outcomes section below. Participants then walked on a treadmill for 3-minutes at their self-selected speed, identified as described above. Finally, participants walked overground along a 10-meter path at their self-selected speed, performing five unidirectional passes. Participants did not interact with the force-field during Assessment sessions.

## Biomechanical data collection

During walking periods of Assessment sessions, participants wore 44 active LED markers (PhaseSpace; San Leandro, CA) placed over their pelvis and lower extremities using a modified Helen Hayes marker set. The location of these markers was sampled at 120 Hz. Our analyses focused on mediolateral motion of the pelvis and feet, as detailed in the Outcomes section below. During Training sessions, a reduced marker set was used to ease setup; markers were placed over the sacrum, heels, and force-field leg cuffs, as done previously when using this force-field [10–12].

## Outcomes

Each intervention group was characterized by several measures collected in the Pre-Intervention Assessment session. Specifically, we recorded participant age, gender, time since stroke, Fugl-Meyer Lower Extremity motor score, Fugl-Meyer Lower Extremity sensory score, and self-reported experience of a fall over the previous year. All clinical tests were performed by a trained, licensed physical therapist or physical therapist assistant. Study outcome measures described below were collected in each Assessment session.

**Paretic $\rho_{SW}$ (primary).**   Using biomechanical data from the treadmill walking trial, we calculated the partial correlation between paretic step width and the mediolateral displacement of the pelvis from the stance foot at the start of the paretic step, accounting for mediolateral

velocity of the pelvis at the start of the step. This metric quantifies the extent to which step width is modulated based on pelvis dynamics, providing insight into this important gait stabilization strategy [3,6]. We have previously found that among PwCS, this metric is significantly reduced for steps taken with the paretic leg [7]. suggesting an altered control of mediolateral stepping behavior.

**Paretic $\rho_{FP}$ (secondary; not pre-planned).** Following the above approach, we calculated the partial correlation between mediolateral paretic foot placement and pelvis displacement at the start of the step, again accounting for pelvis velocity. Here, foot placement is defined as the mediolateral distance from the sacrum to the swing heel when the foot lands from a step. This metric thus focuses on adjustments in swing leg positioning that occur during a step. This metric was added to our outcome measures after study initiation for two reasons. First, we found that our force-field predominantly affects this swing leg modulation, with negligible effects on stance leg motion [11]. Second, we found that this metric (paretic $\rho_{FP}$) was more consistently linked with clinical measures of balance than other biomechanical gait metrics, including paretic $\rho_{SW}$ [8].

**Paretic $\rho_{PD}$ (secondary; not pre-planned).** We additionally calculated the partial correlation between the final mediolateral pelvis displacement at the end of a step and pelvis displacement at the start of the step, again accounting for pelvis velocity. This metric quantifies the extent to which frontal plane stance leg posture upon step onset is related to stance leg posture at the end of the same step, and can likely be partially attributed to the stance leg acting as an inverted pendulum [8]. For the purposes of the present study, this metric essentially provides insight into whether factors other than mediolateral foot placement affect our primary measure of step width modulation. This metric of stance leg position modulation was previously found to not be affected by exposure to our novel force-field [11]. This metric was added to our outcome measures to provide further insight into possible contributors to altered step width modulation.

**Paretic mediolateral foot placement (secondary; not pre-planned).** As described above, paretic mediolateral foot placement was defined as the mediolateral distance from the sacrum to the paretic swing heel when the paretic foot lands at the end of a step. In many PwCS, paretic steps are placed far laterally relative to the pelvis [7,20], which may be a strategy intended to reduce the risk of lateral losses of balance that could occur with overly medial foot placement [21]. This metric was added to our outcome measures after study initiation to determine whether potential increases in foot placement modulation are accompanied by a willingness to walk with more medial foot placement.

**Functional Gait Assessment; FGA (secondary).** This common clinical measure assesses performance of walking balance during tasks of varying difficulty [22].

**Activities-specific Balance Confidence scale; ABC (secondary).** This common clinical scale assesses balance self-efficacy for a range of tasks that may be experienced in the real-world [23].

**Self-selected overground walking speed (secondary).** We calculated overground walking speed from the time required to traverse the middle 6-meters of the 10-meter path, averaged over five passes.

## Self-selected treadmill walking speed (secondary; not pre-planned)

We recorded the treadmill walking speed identified by participants as being the normal speed they would use to walk around the house or the store in each Assessment session. This metric was added to our outcome measures because treadmill speed can affect our primary outcome measure of $\rho_{SW}$ [6].

### Fear of falling (secondary)

Participants were simply asked "Do you have a fear of falling?", which is thought to be a different construct than balance self-efficacy [24].

### Self-reported falls (secondary)

In the Pre-Intervention Assessment session, participants were asked "Have you experienced a fall in which you lost your balance and landed on a lower level in the past year?", following a standard fall definition [25]. Following the first Assessment session, participants were provided with monthly fall calendars in which they marked yes or no to the above question on a daily basis. These calendars were returned to study staff in Assessment sessions.

Additionally, several exploratory tertiary outcomes related to gait biomechanics were assessed (e.g., $\rho_{SW}$, $\rho_{FP}$, $\rho_{PD}$, and average foot placement for steps taken with the non-paretic leg), as described in S1 Appendix.

### Sample size

A total of 54 participants were enrolled into the intervention, with 18 assigned to each intervention group (allocation ratio of 1:1:1). This sample size was based on preliminary results (n = 6), in which exposure to our Perturbing force-field elicited immediate after-effects in which paretic $\rho_{SW}$ increased by 0.18±0.09 (mean±s.d.), whereas exposure to the Assistive force-field elicited no significant after-effects (change of ~0.06). We thus estimated a Cohen's d effect size of ~1.3. With a Bonferroni correction to account for comparisons between intervention groups, we determined that a sample size of 14 participants per group could achieve 80% power. We increased this number to 18 per group to account for potential 20% attrition.

### Randomization sequence

Participants were randomly assigned to one of the three intervention groups using a sequence determined before study onset. The permuted block randomization sequence had a block size of 18, was created by a statistician otherwise not involved with the study procedures, and was stored in a secure online database. A non-blinded study member accessed the group assignment for an individual participant once they completed the Pre-Intervention Assessment session.

### Blinding

Participants were blinded to the intervention group, as all interventions followed the same structure and involved interfacing with the force-field. Clinical assessors were also blinded to the intervention group and were not in the laboratory during training sessions to ensure they did not become aware of the intervention group. Study staff in the laboratory during training sessions were aware of the assigned intervention group, to ensure the force-field was operating correctly. All biomechanical metrics were calculated using standard automated code that did not include group assignment as an input.

### Statistical methods

Our statistical approach was designed with the primary goal of determining whether the outcome measures of interest changed significantly over time within each intervention group, thus addressing our primary and secondary hypotheses. This focus was based on the early stage of the research, as no prior work has investigated whether repeated exposure to this type of mechanical environment has any sustained effects on walking behavior or balance. Our

approach also included an exploratory component, in which changes in outcome measures over time were compared between the Control group and the Perturbing and Assistive groups. Promising results from these analyses would provide justification for a larger-scale future investigation.

Given the early stage of this research, we performed per-protocol analyses that included participants who completed all Assessment sessions. This approach allowed us to focus on the effects in participants who received the prescribed interventions, rather than those whose participation ended due to reasons other than the interventions themselves. However, this approach prevents us from making direct claims regarding clinical effectiveness.

Continuous outcome measures (all those except fear of falling and fall incidence) were analyzed using linear mixed-effects models with repeated measurement, and fitted using restricted maximum likelihood for parameter estimation with Kenward-Roger's adjusted degrees of freedom. Each model included random participant-specific effects on the intercept and slope, with the fixed effects of intervention group, study time point, and their interaction. Time point was used as the repeated measure with first-order autoregressive AR(1) covariance structure. Statistical tests were conducted as two-tailed tests with alpha value set to 0.05 for all comparisons.

Assumptions of residuals were first assessed to determine whether each outcome measure exhibited non-normality or heteroscedasticity. While three outcome measures (ABC, paretic $\rho_{PD}$, non-paretic $\rho_{PD}$) exhibited outlier values that skewed the distribution, the distributions were not improved by common transformation techniques, and were thus modeled using their raw data values under a Gaussian distribution with identity link. For interpretation of the model results, we focused on three sets of comparisons. First, we assessed whether each outcome measure changed across Assessment sessions (time points) for each intervention group (Control, Assistive, and Perturbing), addressing our primary and secondary hypotheses. We then assessed whether each outcome measure differed from the Control group at each time point (for the Assistive and Perturbing groups). Finally, we assessed whether the change in each outcome measure from its baseline value differed from the Control group at each subsequent time point (for the Assistive and Perturbing groups), addressing our exploratory questions. Fear of falling and fall incidence were analyzed using generalized linear mixed-effects models with AR(1) covariance structure for repeated measures, modeled using a binary distribution with logit link and a negative binomial distribution with log link function, respectively. Fall incidence results were presented as incidence rate (IR) and IR ratios (IRR) with 95% confidence intervals (CI). Statistical analyses were conducted in SAS v9.4 (SAS Institute Inc., Cary, NC), by means of PROC GLIMMIX.

The statistical methods described above are an expansion of our originally planned statistical approach, which simply involved repeated measures ANOVAs with the dependent variable of the change relative to baseline for each outcome measure, and the independent variable of intervention group (Control, Assistive, and Perturbing), and time point (Week 4, Week 8, Week 12, and Follow-up). Upon consultation with the study statistician, we revised our approach to that described above, which allowed us to account for the different baseline values in our primary outcome measure and make more detailed comparisons at individual time points. For transparency, the results of our originally planned statistical analyses are presented in S2 Appendix.

## Results

### Recruitment and participant flow

Study recruitment occurred between February 2017 and October 2019. A total of 93 participants were recruited into the study and completed an initial screening session in which their

**Table 1. Participant characteristics.**

| Participant characteristics | Control (n = 13) | Assistive (n = 14) | Perturbing (n = 17) |
|---|---|---|---|
| Gender (female/male) | 6F / 7M | 7F / 7M | 6F / 11M |
| Affected side (left/right) | 7L / 6R | 8L / 6R | 8L / 9R |
| Age (yrs) | 55 [23–77] | 63.5 [46–85] | 61 [35–76] |
| Height (cm) | 175 [155–187] | 167 [151–188] | 171 [152–196] |
| Mass (kg) | 79 [59–111] | 81 [60–127] | 84 [59–143] |
| Time since stroke (mos) | 25 [11–242] | 41.5 [6–199] | 39 [6–161] |
| Fugl-Meyer Lower Extremity motor score | 25.5 [17–34] | 24 [16–34] | 21 [13–33] |
| Fugl-Meyer Lower Extremity sensory score | 10 [2–12] | 12 [4–12] | 10 [1–12] |
| Fall in prior year (yes/no) | 7Y / 6N | 6Y / 8N | 7Y / 10N |

Participant characteristics from the Pre-Intervention session for the three intervention groups. Continuous measures are presented as median [range].

eligibility to participate was assessed. Recruitment was stopped once 54 eligible individuals agreed to participate in the trial and completed the Pre-Intervention Assessment session. Participant flow is illustrated in Fig 1. The trial ended in March 2020, when research operations at the Medical University of South Carolina were stopped in response to the COVID-19 pandemic. At that time, the final three participants were enrolled in the intervention. The total attrition rate from randomization to follow-up was 19% (13% before the research shutdown). The attrition rate was 28% in the Control group, 22% in the Assistive group, and 6% in the Perturbing group. Summary participant characteristics collected in the Pre-Intervention session for the three intervention groups are presented in Table 1.

## Intervention adherence

Across participants who completed the study, the median number of attended training visits was 23 (range = 18–24) for the Control group, 22 (range = 20–24) for the Assistive group, and 23 (range = 19–24) for the Perturbing group.

## Study outcomes

All primary and secondary study outcome measure values are provided as individual level data in S3 Appendix and group level data in S4 Appendix, and are presented graphically in the figures below, with accompanying statistical results in the following tables.

## Primary biomechanical gait outcome

The relationship between paretic step width and mediolateral pelvis displacement (paretic $\rho_{SW}$) only changed significantly for the Perturbing group (Fig 3A and 3E; Table 2). Specifically, paretic $\rho_{SW}$ in the Perturbing group increased significantly from its Pre-Intervention value for all subsequent time points ($p \leq 0.017$). In contrast, paretic $\rho_{SW}$ did not change significantly from its Pre-Intervention value at any subsequent time point in the Control group ($p \geq 0.48$) or the Assistive group ($p \geq 0.12$). From our exploratory analyses, in comparison to the Control group, the Perturbing group had a lower baseline paretic $\rho_{SW}$ value ($p = 0.026$) but exhibited larger increases in this metric at Week 4 ($p = 0.0097$), Week 8 ($p = 0.021$), and Follow-up ($p = 0.044$).

## Secondary biomechanical gait outcomes

Across our secondary biomechanical outcomes, the only consistent changes across assessments were observed in the Perturbing group for paretic foot placement modulation (paretic

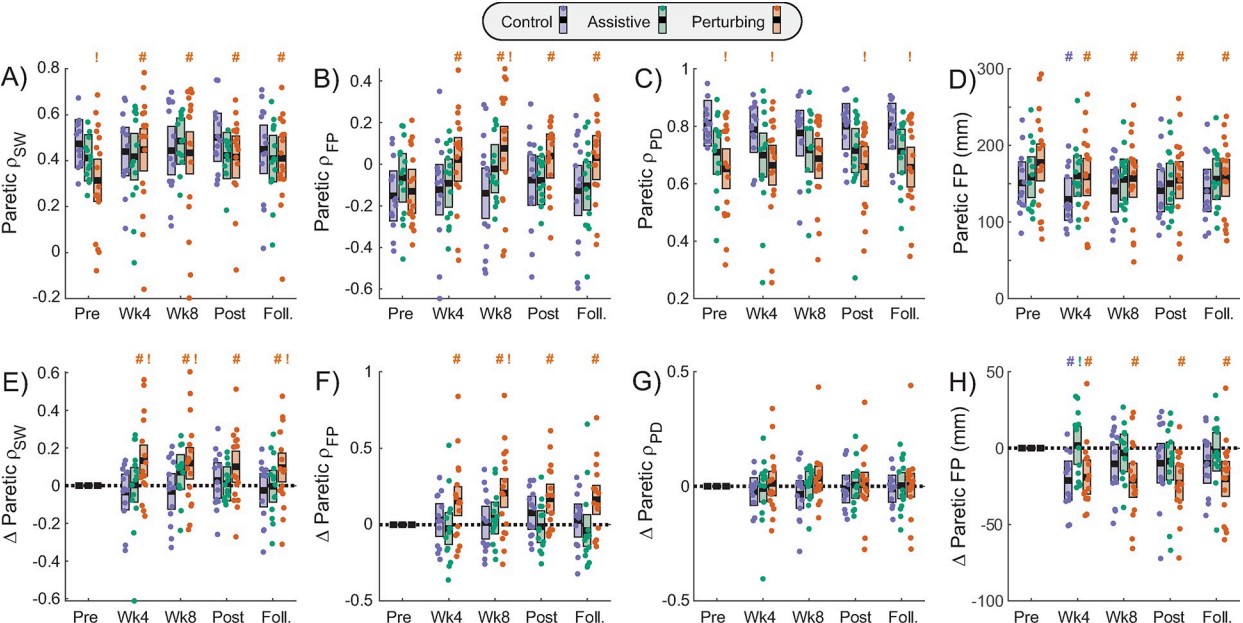

**Fig 3. Biomechanical gait outcome measures.** Metrics are presented for each assessment time point, with the top row (A-D) illustrating raw metric values, and the bottom row (E-H) illustrating changes from each participant's Pre-Intervention value. The five assessment time points are Pre-Intervention (Pre), after 4-weeks of training (Wk4), after 8-weeks of training (Wk8), Post-Intervention (Post), and after a 12-week Follow-Up period (Foll.). Thick black lines indicate each intervention group's model-estimated mean value at each time point, shaded areas indicate 95% confidence intervals, and dots indicate individual participant values. Pound signs (#) indicate a significant difference from the group's Pre-Intervention metric value, while exclamation marks (!) indicate a significant difference from the Control group value at the marked time point. Statistical indicators are color coded to match the corresponding group.

$\rho_{FP}$) (Fig 3B and 3F; Table 2) and average paretic foot placement (Fig 3D and 3H; Table 2). For the Perturbing group, paretic $\rho_{FP}$ increased relative to its Pre-Intervention value at all subsequent time points (p≤0.0015), and the average paretic foot placement decreased (became more medial) relative to the Pre-Intervention value at all later time points (p≤0.001). In contrast, paretic $\rho_{FP}$ did not change significantly over time for either the Control group (p≥0.17) or the Assistive group (p≥0.41). Average paretic foot placement decreased significantly only at the Week 4 timepoint for the Control group, and did not change across assessments for the Assistive group (p≥0.18). Final pelvis displacement modulation during paretic steps (paretic $\rho_{PD}$) did not vary significantly across assessment sessions in the Control group (p≥0.26), the Assistive group (p≥0.74), or the Perturbing group (p≥0.18) (Fig 3C and 3G; Table 2). From our exploratory analyses, compared to the Control group, the Perturbing group exhibited significantly larger paretic $\rho_{FP}$ values and significantly greater increases in paretic $\rho_{FP}$ at the Week 8 time point (p≤0.009). Final pelvis displacement modulation during paretic steps (paretic $\rho_{PD}$) was significantly lower in the Perturbing group than the Control group at most time points.

## Secondary clinical outcomes

Several of the assessed clinical outcomes changed significantly across the intervention duration, although no significant differences between groups were observed in our exploratory analyses. FGA score generally increased over the duration of the intervention (Fig 4A and 4E; Table 3), with significant improvements relative to the Pre-Intervention value at every subsequent time point in the Control (p≤0.016) and Perturbing (p≤0.013) groups, but less consistent improvements in the Assistive group. In contrast, ABC score only increased significantly

**Table 2. Biomechanical gait outcome measures.**

| Outcome Metrics | Comparisons to baseline within each intervention group | | | Comparisons of metric values with Control group | | Comparison of metric changes with Control group | |
|---|---|---|---|---|---|---|---|
| | Control | Assistive | Perturbing | Assistive | Perturbing | Assistive | Perturbing |
| | t (p) | t (p) | t (p) | t (p) | t (p) | t (p) | t (p) |
| **Primary biomechanical gait outcome:** | | | | | | | |
| **Paretic $\rho_{SW}$** | | | | | | | |
| Baseline | | | | -0.84 (0.40) | **-2.28 (0.026)** | | |
| Week 4 | -0.71 (0.48) | 0.10 (0.92) | **3.17 (0.0019)** | -0.31 (0.76) | 0.11 (0.91) | 0.58 (0.56) | **2.63 (0.0097)** |
| Week 8 | -0.63 (0.53) | 1.57 (0.12) | **2.84 (0.0053)** | 0.56 (0.57) | -0.15 (0.88) | 1.54 (0.13) | **2.34 (0.021)** |
| Week 12 | 0.58 (0.56) | 0.22 (0.83) | **2.42 (0.017)** | -1.09 (0.28) | -1.23 (0.22) | -0.27 (0.79) | 1.15 (0.25) |
| Follow-up | -0.55 (0.58) | -0.07 (0.94) | **2.47 (0.015)** | -0.55 (0.59) | -0.55 (0.58) | 0.35 (0.73) | **2.04 (0.044)** |
| **Secondary biomechanical gait outcomes:** | | | | | | | |
| **Paretic $\rho_{FP}$** | | | | | | | |
| Baseline | | | | 1.04 (0.30) | 0.27 (0.79) | | |
| Week 4 | 0.55 (0.58) | -0.49 (0.62) | **3.23 (0.0015)** | 0.37 (0.71) | 1.81 (0.07) | -0.74 (0.46) | 1.71 (0.09) |
| Week 8 | 0.24 (0.81) | 0.83 (0.41) | **4.34 (<0.0001)** | 1.40 (0.17) | **2.67 (0.0090)** | 0.40 (0.69) | **2.67 (0.0083)** |
| Week 12 | 1.38 (0.17) | -0.25 (0.80) | **3.53 (0.0005)** | -0.02 (0.98) | 1.43 (0.16) | -1.17 (0.24) | 1.28 (0.20) |
| Follow-up | 0.47 (0.64) | -0.75 (0.46) | **3.40 (0.0009)** | 0.27 (0.79) | 1.97 (0.05) | -0.86 (0.39) | 1.89 (0.06) |
| **Paretic $\rho_{PD}$** | | | | | | | |
| Baseline | | | | -1.85 (0.07) | **-3.02 (0.0036)** | | |
| Week 4 | -0.75 (0.46) | -0.33 (0.74) | 0.47 (0.64) | -1.61 (0.11) | **-2.35 (0.022)** | 0.31 (0.76) | 0.87 (0.38) |
| Week 8 | -1.13 (0.26) | 0.27 (0.78) | 1.35 (0.18) | -1.07 (0.29) | -1.69 (0.10) | 1.01 (0.32) | 1.74 (0.08) |
| Week 12 | -0.36 (0.72) | 0.20 (0.84) | 0.30 (0.77) | -1.54 (0.13) | **-2.66 (0.0097)** | 0.40 (0.69) | 0.47 (0.64) |
| Follow-up | -0.33 (0.74) | 0.12 (0.90) | 0.22 (0.82) | -1.60 (0.11) | **-2.72 (0.0083)** | 0.33 (0.74) | 0.40 (0.69) |
| **Paretic mediolateral foot placement** | | | | | | | |
| Baseline | | | | 0.40 (0.69) | 1.47 (0.15) | | |
| Week 4 | **-3.28 (0.0013)** | 0.27 (0.79) | **-3.35 (0.001)** | 1.60 (0.12) | 1.59 (0.12) | **2.55 (0.012)** | 0.26 (0.80) |
| Week 8 | -1.59 (0.11) | -0.50 (0.62) | **-3.72 (0.0003)** | 0.78 (0.44) | 0.88 (0.38) | 0.80 (0.43) | -1.25 (0.21) |
| Week 12 | -1.51 (0.13) | -1.36 (0.18) | **-4.03 (<0.0001)** | 0.47 (0.64) | 0.76 (0.45) | 0.14 (0.89) | -1.51 (0.13) |
| Follow-up | -1.49 (0.14) | -0.38 (0.71) | **-3.46 (0.0007)** | 0.79 (0.43) | 0.91 (0.37) | 0.81 (0.42) | -1.16 (0.25) |

Statistical results are presented for biomechanical gait outcome measures. Bolding indicates a significant effect (p<0.05).

from the Pre-Intervention value at the Post-Intervention and Follow-up time points in the Assistive group (p≤0.046) (Fig 4B and 4F; Table 3). Significant increases in self-selected over-ground walking speed occurred at the Post-Intervention time point for the Control group (p = 0.020) and the Assistive group (p = 0.048), but not the Perturbing group (p = 0.32) (Fig 4C and 4G; Table 3). Self-selected treadmill walking speed increased significantly relative to its Pre-Intervention value at all time points for the Control group (p≤0.001) and Perturbing group (p≤0.026), and at Week 8 and Week 12 for the Assistive group (p≤0.0044) (Fig 4D and 4H; Table 3).

The probability of self-reported fear of falling ranged from 29% to 50% across intervention groups and time points (Fig 5A; Table 3) but did not differ significantly across time points for any group (p≥0.29). Across the entire duration of the 12-week intervention and 12-week follow-up period, the fall incidence rate was 1.23 (falls/person) for the Control group, 0.93 for the Assistive group, and 0.65 for the Perturbing group (Fig 5B and 5C; Table 4). The fall incidence rate in the Perturbing group across the full duration of the study was significantly (p = 0.0053) lower than in the Control group. While a similar pattern of approximately half as many falls in

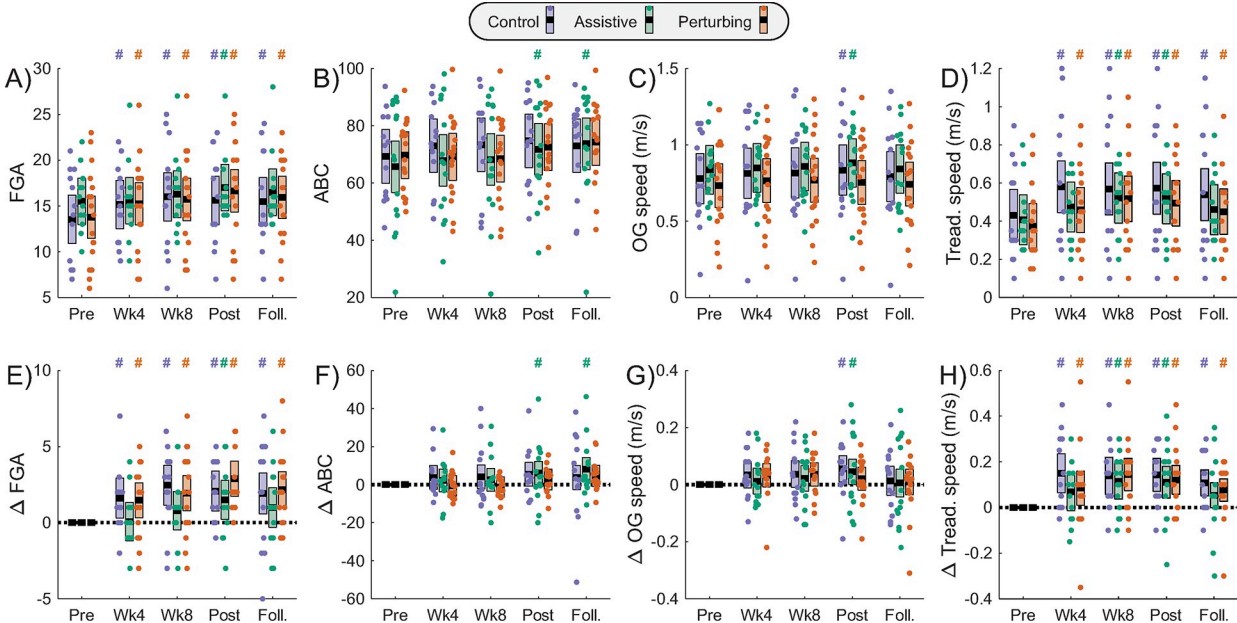

**Fig 4. Continuous clinical outcome measures.** Metrics are presented for each assessment time point, following the same structure as in Fig 3.

the Perturbing group than the Control group was observed when isolating fall incidence rate during the intervention or the follow-up period, no significant differences between groups were detected (p≥0.24).

## Harms

While many participants experienced falls during daily life, no falls occurred during study procedures. One participant experienced notable muscle soreness after a study visit, which was probably related to study participation. Two participants experienced negative health events that were possibly related to study participation: one reported severe fatigue following a training session and a second reported severe dizziness at the start of a training session. Following medical visits, these participants were diagnosed with low hemoglobin levels and inflamed facial nerves, respectively. Four participants experienced negative health events that were unrelated to study participation: one experienced a stroke at home while enrolled in the study, one experienced stroke-like symptoms that were attributed to recrudescence, one experienced a broken wrist from a car accident, and one was diagnosed with pneumonia.

## Discussion

Mediolateral step modulation, an important gait stabilization strategy, is disrupted for paretic steps in many PwCS. In general, we found that this strategy was strengthened with repeated exposure to perturbations, but not with exposure to assistance or a control condition. The study results only partially support our hypotheses, as paretic step width modulation, mediolateral paretic foot placement modulation, and FGA score all increased relative to baseline in the Perturbing intervention group. However, statistically significant differences between the Perturbing and Control groups were not consistently detected, and no significant changes in ABC score or fear of falling were observed in the Perturbing group. The Perturbing group experienced a lower rate of real-world falls than the Control group, although this result should be interpreted with caution.

**Table 3. Clinical outcome measures.**

| Outcome Metrics | Comparisons to baseline within each intervention group | | | Comparisons of metric values with Control group | | Comparison of metric changes with Control group | |
|---|---|---|---|---|---|---|---|
| | **Control** | **Assistive** | **Perturbing** | **Assistive** | **Perturbing** | **Assistive** | **Perturbing** |
| | t (p) | t (p) | t (p) | t (p) | t (p) | t (p) | t (p) |
| **Secondary clinical outcomes:** | | | | | | | |
| **FGA** | | | | | | | |
| Baseline | | | | 1.07 (0.29) | 0.13 (0.90) | | |
| Week 4 | **2.43 (0.016)** | 0.11 (0.91) | **2.53 (0.013)** | 0.23 (0.82) | 0.05 (0.96) | -1.67 (0.10) | -0.16 (0.87) |
| Week 8 | **3.70 (0.0003)** | 1.23 (0.22) | **3.34 (0.0011)** | 0.16 (0.88) | -0.17 (0.87) | -1.81 (0.07) | -0.59 (0.56) |
| Week 12 | **3.11 (0.0023)** | **2.33 (0.021)** | **4.93 (<0.0001)** | 0.76 (0.45) | 0.59 (0.56) | -0.62 (0.54) | 0.91 (0.37) |
| Follow-up | **2.82 (0.0056)** | 1.52 (0.13) | **3.65 (0.0004)** | 0.57 (0.57) | 0.27 (0.78) | -0.97 (0.33) | 0.28 (0.78) |
| **ABC** | | | | | | | |
| Baseline | | | | -0.56 (0.58) | 0.08 (0.94) | | |
| Week 4 | 1.19 (0.24) | 0.72 (0.47) | -0.23 (0.82) | -0.80 (0.43) | -0.63 (0.53) | -0.35 (0.72) | -1.05 (0.30) |
| Week 8 | 1.30 (0.20) | 0.84 (0.40) | -0.45 (0.65) | -0.80 (0.43) | -0.78 (0.44) | -0.36 (0.72) | -1.28 (0.20) |
| Week 12 | 1.73 (0.09) | **2.02 (0.046)** | 1.00 (0.32) | -0.45 (0.65) | -0.36 (0.72) | 0.15 (0.88) | -0.64 (0.52) |
| Follow-up | 1.16 (0.25) | **2.61 (0.010)** | 1.64 (0.10) | 0.11 (0.91) | 0.22 (0.83) | 0.97 (0.33) | 0.20 (0.84) |
| **Overground speed** | | | | | | | |
| Baseline | | | | 0.50 (0.62) | -0.43 (0.67) | | |
| Week 4 | 1.46 (0.15) | 0.53 (0.60) | 1.59 (0.11) | 0.31 (0.76) | -0.44 (0.66) | -0.68 (0.50) | -0.05 (0.96) |
| Week 8 | 1.56 (0.12) | 0.97 (0.33) | 1.74 (0.08) | 0.37 (0.71) | -0.44 (0.66) | -0.45 (0.65) | -0.03 (0.97) |
| Week 12 | **2.36 (0.020)** | **1.99 (0.048)** | 1.03 (0.30) | 0.41 (0.68) | -0.74 (0.46) | -0.32 (0.75) | -1.10 (0.27) |
| Follow-up | 0.51 (0.61) | 0.23 (0.82) | 0.42 (0.68) | 0.44 (0.66) | -0.46 (0.65) | -0.21 (0.84) | -0.11 (0.91) |
| **Treadmill speed** | | | | | | | |
| Baseline | | | | -0.25 (0.80) | -0.64 (0.53) | | |
| Week 4 | **3.48 (0.0008)** | 1.64 (0.11) | **2.27 (0.026)** | -1.13 (0.27) | -1.36 (0.18) | -1.37 (0.17) | -1.13 (0.26) |
| Week 8 | **3.40 (0.0010)** | **2.91 (0.0044)** | **4.05 (< .0001)** | -0.51 (0.61) | -0.57 (0.57) | -0.43 (0.67) | 0.10 (0.92) |
| Week 12 | **3.89 (0.0001)** | **3.14 (0.0020)** | **3.77 (0.0002)** | -0.59 (0.56) | -0.88 (0.38) | -0.62 (0.53) | -0.45 (0.66) |
| Follow-up | **3.77 (0.0002)** | 1.94 (0.05) | **3.06 (0.0026)** | -0.83 (0.41) | -0.99 (0.33) | -1.36 (0.18) | -0.82 (0.41) |
| **Fear of falling** | | | | | | | |
| Baseline | | | | -0.14 (0.89) | 0.15 (0.89) | | |
| Week 4 | 0 (>0.99) | 0.90 (0.37) | 0 (>0.99) | 0.58 (0.56) | 0.15 (0.89) | 0.62 (0.53) | 0 (>0.99) |
| Week 8 | -0.53 (0.60) | -0.52 (0.60) | -0.46 (0.65) | -0.12 (0.90) | 0.25 (0.80) | 0.01 (0.99) | 0.11 (0.91) |
| Week 12 | -0.61 (0.54) | -0.60 (0.55) | -1.06 (0.29) | -0.12 (0.90) | -0.08 (0.94) | 0.02 (0.99) | -0.24 (0.81) |
| Follow-up | -0.81 (0.42) | 0 (>0.99) | 0 (>0.99) | 0.26 (0.79) | 0.56 (0.57) | 0.59 (0.56) | 0.62 (0.54) |

Statistical results are presented for clinical outcome measures. Bolding indicates a significant effect (p<0.05).

**Table 4. Falls outcome measure.**

| Outcome Metrics | Comparisons of metric values with Control group | |
|---|---|---|
| | **Assistive** | **Perturbing** |
| | t (p) | t (p) |
| **Fall incidence** | | |
| Total | -1.25 (0.21) | **-2.82 (0.0053)** |
| During intervention | -0.56 (0.58) | -0.90 (0.37) |
| During follow-up | -0.32 (0.75) | -1.20 (0.24) |

Statistical results are presented for the incidence rate of real-world falls. Bolding indicates a significant effect (p<0.05).

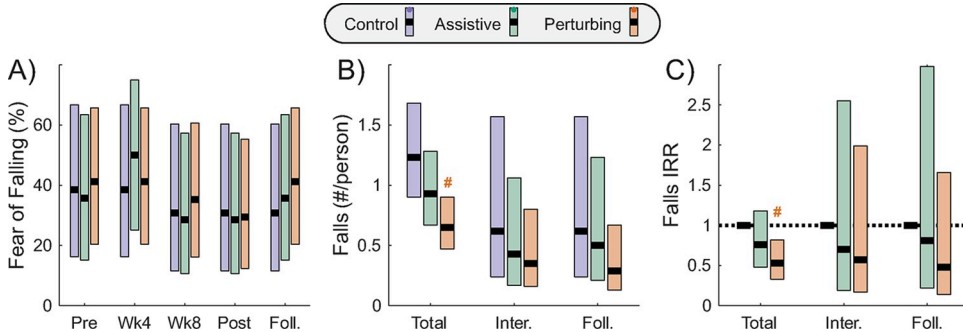

**Fig 5. Dichotomous and discrete measures related to falls.** Metrics are presented for the intervention and follow-up time periods. A) The percentage of participants who reported a fear of falling is presented at each assessment time point. B) The incidence rate for falls experienced by participants in each intervention group is presented for the entire 24-week period (Total), as well as during the 12-week intervention (Inter.) and the 12-week follow-up period (Foll.). C) The incidence rate ratio in comparison to the Control group is presented for the Assistive and Perturbing group for the same three time windows. Thick black lines indicate each intervention group's mean value, shaded areas indicate 95% confidence intervals, and pound signs (#) indicate a significant difference from the Control group in the corresponding time window.

Significant changes in biomechanical gait outcomes were only observed in the Perturbing group. Repeated exposure to targeted perturbations was followed by sustained increases in paretic step width modulation ($\rho_{SW}$), indicating improved use of this important gait stabilization strategy [3]. These changes can be attributed to increased foot placement modulation ($\rho_{FP}$) rather than stance leg modulation ($\rho_{PD}$), likely because our force-field primarily targets swing leg positioning during a step [12]. In contrast, the Assistance and Control groups did not exhibit notable changes in these metrics. These results are consistent with our prior work investigating the effects of our force-field within a single session; in both neurologically-intact controls [10,11] and PwCS [12], after-effects with increased step width modulation were observed following perturbations, but not assistance. However, our exploratory statistical comparisons did not consistently support the stronger claim that the Perturbing group exhibited larger effects than the Control group, likely due in part to the higher-than-expected attrition rate in the Control group (28%). Additionally, the power analysis used to choose the sample size for this study was based on the effect size of the increases in paretic $\rho_{SW}$ that *immediately* followed exposure to force-field perturbations (Cohen's d of 1.3 in initial preliminary data, and 1.0 based on subsequent published work [12]). In contrast, repeated exposure to perturbations here produced a notably smaller *sustained* effect size of 0.6 (across time points), reducing the likelihood of detecting significant effects. Finally, the average baseline value of paretic $\rho_{SW}$ was notably lower in the Perturbing group than the Control group, despite participant randomization. This baseline difference may have made it more likely to observe an increase over time in the Perturbing group due simply to regression to the mean. To explore this possibility,

**Table 5. Primary outcome measure with baseline outliers removed.**

|  | Control | Assistive | Perturbing |
|---|---|---|---|
| Baseline: mean [95% CI] | 0.47 [0.38 0.57] | 0.41 [0.34 0.49] | 0.41 [0.34 0.49] |
| Change: mean [95% CI] | -0.015 [-0.080 0.050] | 0.021 [-0.053 0.094] | **0.091 [0.026 0.156]** |
| Change: t (p) | -0.47 (0.64) | 0.58 (0.57) | **2.81 (0.007)** |

Statistical results are presented for paretic $\rho_{SW}$ measures after four outliers from the Perturbing group were removed. Bolding indicates a significant effect (p<0.05).

we performed a post-hoc analysis in which we removed the four participants in the Perturbing group with low outlier values for the baseline paretic $\rho_{SW}$ metric (see Fig 3A). Following the general statistical approach described in the Methods, we used linear mixed effect models to test whether each intervention group exhibited a significant increase in paretic $\rho_{SW}$, focused on a comparison between baseline and all later timepoints, given the lack of apparent changes from Week 4 to Follow-Up. Here, the baseline values of paretic $\rho_{SW}$ were similar across groups, and only the Perturbing group exhibited a significant increase in this metric at later timepoints (Table 5). Therefore, the change in paretic $\rho_{SW}$ for the Perturbing group does not appear to be solely due to having a lower baseline value. This concern is further ameliorated by our observation of a similar pattern of improvement only in the Perturbing group for paretic $\rho_{FP}$, a metric for which no baseline differences between groups were present.

The changes in paretic step modulation observed solely in the Perturbing group are consistent with the mechanism of error-driven sensorimotor learning. During human movement, sensory prediction errors occur when sensory feedback does not match the expected feedback for that task [13]. Experiments with novel sensorimotor environments that increase these errors (e.g., reaching to a target with altered visual feedback, walking on a split-belt treadmill) have revealed both gradual changes in the movement pattern within this environment and short-lived after-effects when the environment returns to its normal state [26,27], indicative of sensorimotor adaptation in which the underlying control strategy is adjusted. In concept, repeated exposure to a novel environment could elicit sensorimotor learning, through which targeted movement characteristics are sustained in normal movement contexts [28]. In the present work, perturbations may have increased the perceived foot placement errors, motivating participants to adjust their control strategy to prevent a loss of balance. In contrast, neither assistance nor simply walking would be expected to increase errors and promote a change in strategy. In order for the initially short-lived strategy changes to be retained over long time periods, the new movement pattern presumably must be perceived as superior in some way, or individuals would be expected to simply adapt back to their original pattern [13]. Speculatively, perhaps the baseline gait pattern with limited paretic step modulation was a habitual behavior learned in the early post-stroke period. A reduced ability to accurately control motion of the paretic swing leg may encourage a strategy in which the paretic foot is consistently placed quite lateral from the pelvis in order to prevent lateral losses of balance during the subsequent single stance phase [21], despite the accompanying increased mechanical demands [29,30]. Repeated periods in which the delivered perturbations elicited greater active step modulation may have prompted participants to recognize that this was a gait pattern that could be less demanding and more stable, promoting its continued use. The accompanying shift in average paretic foot placement to a less lateral location is consistent with this possibility.

While error-driven learning thus appears to be a reasonable explanation for our results, little prior evidence exists for leveraging this mechanism to drive targeted gait changes in PwCS. The classic example of using split-belt walking to augment errors has indeed reduced post-stroke step length asymmetries with repeated exposure [15,16], although similar results have been achieved by minimizing errors or simply providing verbal feedback [31]. Additionally, repeated exposure to leg swing resistance designed to increase step length errors had very similar effects to leg swing assistance on the gait characteristics of PwCS [32]. One possible explanation for the unique role of perturbations in the present study is the important role of perceived imbalance in eliciting locomotor adaptation [33]. Perhaps only the Perturbing group perceived the force-field as challenging their balance and changed their gait pattern accordingly, while the Assistive and Control groups had no such motivation to adapt. In contrast, post-stroke walking balance appears to be challenged not just by augmenting step length asymmetry, but also by interventions that *reduce* step length asymmetry [34]. The threat of a

loss of balance may be an important component of encouraging an individual to change their habitual walking pattern.

The unique beneficial effects of perturbations were less apparent for clinical outcomes than for the biomechanical gait outcomes. FGA score and self-selected treadmill walking speed tended to increase over time in all intervention groups, possibly due to a practice effect as participants repeatedly performed these initially unfamiliar tasks [35]. None of the other clinical assessments exhibited notable improvements in the Perturbing group. The lack of clear unique perturbation effects on our clinical outcomes, despite changes in gait mechanics, may be due in part to the only moderate to weak relationships between such measures [8]. Many factors other than the ability to modulate swing foot location are likely to influence clinical measures of balance and confidence.

While participants in the Perturbing group had a fall incidence rate of approximately half that of participants in the Control group, this early-stage result should be viewed skeptically. Although retrospectively quantifying self-reported falls without the use of fall calendars is notoriously unreliable [25], a lower proportion of Perturbing group participants (41%) reported experiencing a fall in the prior year than Control group participants (54%), so baseline fall risk may have differed between these groups irrespective of any intervention. Additionally, focusing analyses solely on the follow-up period (which presumably would be most relevant for clinical implementation) did not reveal any significant differences between groups, likely due in part to the study's small sample size. Finally, numerous promising interventions for improving balance have not produced significant decreases in real-world fall rate among PwCS once progressed to large-scale clinical trials [36–39]. The present results would need to be replicated on a larger scale before implementation of this type of perturbation for the purposes of fall prevention.

The type of perturbations delivered in this study are just one option of many, each with potential distinct effects on post-stroke balance. The goal of the present perturbations was to retrain a proactive balance strategy, in which individuals actively adjust where their foot is placed with each step [40]. In part, this approach was based on the idea that many post-stroke falls are attributable to "intrinsic" factors, whereby falls are caused by movement errors rather than external perturbations [41]. In contrast, most perturbation methods are designed to elicit reactive responses distinct from the steady-state walking pattern. One well-studied type of reactive perturbation training focuses on standing balance, in which participants must take step responses to recover from an external perturbation [42]. This type of training can successfully improve post-stroke reactive responses [43,44]. Other perturbation types, including lateral push perturbations during standing [45] and walking [46], trips [47], and slips [48] have all generally revealed improved reactive responses in a variety of clinical populations, as summarized in a recent review [49]. It is presently unclear whether these types of reactive perturbation training prevent subsequent real-world falls in PwCS [39,50]. It is also unclear whether the apparent specificity of perturbation training effects [51] (e.g., slip training improves slip responses but not trip responses) will ultimately require perturbation training to include a wide range of perturbation types, perhaps both proactive and reactive.

A limitation of this work is the focus on a single biomechanical gait strategy that contributes to straight-line, steady-state walking balance. While adjustments in mediolateral foot placement location are generally accepted to be a primary method of ensuring side-to-side balance, other strategies can also contribute, including shifting the center of pressure location under the stance foot [52,53], and adjusting trail leg push-off magnitude [53]. It is unclear whether improvements in step modulation during treadmill walking translate to the more challenging contexts experienced when walking overground, including turning and changing speeds [54]. Each of these factors may have contributed to the lack of clear effects of our perturbation

methods on clinical outcomes that involve more complexity. However, the present results provide support for applying novel mechanical environments to promote sustained changes in post-stroke gait patterns. Strategies other than foot placement modulation could conceivably be promoted using similar targeted approaches. For example, deficits in controlling motion of the center of mass relative to the stance foot could be targeted using repeated exposure to controlled forces applied to the pelvis [46,55–58], while deficits in push-off modulation could be targeted through repeated exposure to controlled ankle moments [59].

In conclusion, the present work contributes to the evidence that post-stroke gait patterns can be modified through appropriately targeted perturbations. However, several important questions remain unanswered. It is unclear whether the modified gait pattern with greater step modulation should be considered preferable to the original gait pattern, as evidence of clinical improvements is elusive. It is also unclear whether proactive perturbation methods and reactive perturbation methods targeting the same gait behavior would have differential effects, or whether either method can convincingly reduce the rate of real-world falls in PwCS, an ultimate goal of much of this work.

## Supporting information

**S1 Checklist. CONSORT 2010 checklist of information to include when reporting a randomised trial\*.**
(DOCX)

**S1 Appendix. Exploratory tertiary outcomes related to gait biomechanics.**
(DOCX)

**S2 Appendix. Results of originally planned statistical analyses.**
(DOCX)

**S3 Appendix. Individual level data.**
(XLSX)

**S4 Appendix. Group level data summary.**
(DOCX)

**S1 File.**
(DOCX)

## Author Contributions

**Conceptualization:** Jesse C. Dean.

**Data curation:** Andrea D. Boan.

**Formal analysis:** Nicholas K. Reimold, Andrea D. Boan.

**Funding acquisition:** Jesse C. Dean.

**Investigation:** Alexa A. Krause, Nicholas K. Reimold, Aaron E. Embry, Camden J. Jacobs, Jesse C. Dean.

**Methodology:** Heather L. Knight.

**Project administration:** Jesse C. Dean.

**Software:** Nicholas K. Reimold, Heather L. Knight.

**Supervision:** Jesse C. Dean.

**Visualization:** Jesse C. Dean.

**Writing – original draft:** Jesse C. Dean.

**Writing – review & editing:** Alexa A. Krause, Nicholas K. Reimold, Aaron E. Embry, Heather L. Knight, Camden J. Jacobs, Andrea D. Boan, Jesse C. Dean.

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
