## [Decision Letter · Decision Letter 0]

8 Apr 2024

PONE-D-24-04911Effect of mediolateral leg perturbations on walking balance in people with chronic stroke: a randomized controlled trialPLOS ONE

Dear Dr. Dean,

Thank you for submitting your manuscript to PLOS ONE. After careful consideration, we feel that it has merit but does not fully meet PLOS ONE’s publication criteria as it currently stands. Therefore, we invite you to submit a revised version of the manuscript that addresses the points raised during the review process.

We look forward to receiving your revised manuscript.

Kind regards,

Ryan T. Roemmich

Academic Editor

PLOS ONE

Journal Requirements:

**Additional Editor Comments:**

Thank you for submitting your work to PLOS ONE. It has now been reviewed by three independent reviewers who all largely felt positively about the work. They have provided a series of comments and suggestions; please be sure to address these via point-by-point responses in your resubmission. We look forward to receiving your revised manuscript.

Reviewers' comments:

Reviewer's Responses to Questions

**Comments to the Author**

1. Is the manuscript technically sound, and do the data support the conclusions?

Reviewer #1: Partly

Reviewer #2: Yes

Reviewer #3: Yes

2. Has the statistical analysis been performed appropriately and rigorously? 

Reviewer #1: No

Reviewer #2: Yes

Reviewer #3: I Don't Know

3. Have the authors made all data underlying the findings in their manuscript fully available?

Reviewer #1: Yes

Reviewer #2: Yes

Reviewer #3: Yes

4. Is the manuscript presented in an intelligible fashion and written in standard English?

Reviewer #1: Yes

Reviewer #2: Yes

Reviewer #3: Yes

5. Review Comments to the Author

Reviewer #1: The purpose of this study was to compare the effects of walking training with no forces vs. assistive forces vs. perturbing forces on step width modulation in individuals with chronic stroke. They found that the perturbing forces tended to yield changes in biomechanical outcomes, whereas this was less common in the other two groups. Overall, the manuscript is well-written with good rationale for the approach. The methods employed also appear to be appropriate for the study. I have some concerns, with the most pressing being some limitations of the statistical approach. My major and minor concerns are listed.

Major comments

• The primary weakness I see is that the statistical methods are somewhat lacking. I appreciate the transparency about the original analysis plan and the reasons it was changed. However, the approach used (unless I misunderstand) does not provide insight into any group X time interactions. On Page 14, Line 283-286, you state that the analysis focused on within-group changes across time and on between group comparisons at each time point. This will not provide insight into whether one intervention led to greater changes in your outcomes than another. However, the results imply that you are providing insight into interactions (e.g., Page 17, line 344-345, “exhibited larger increases.” So, first, please clarify whether your statistical approach looked at group X time interactions or compared change values at each time point. If the statistical design was not designed in either of these ways, the results and discussion should not be focused on interactions. For example, if the statistical approach is as described on Page 14, the presentation of change values in Table 2 is misleading. It makes it seem that the statistical tests show a greater change in the Perturbing group, when in fact, this was not tested.

• My guess is that you did not have enough power to test for interaction effects for several reasons. 1) your sample size calculation was based on 2, not 3 groups. 2) the large number of timepoints included in the model reduced the number of degrees of freedom. This may be why you don’t state a hypothesis about one intervention compared to another despite the introduction clearly implying that the error augmentation group will have greater improvements/changes. I think the best solution to this problem would be to revise the statistical approach to have two timepoints (Pre/Post) or maybe 3 (Pre/Post/FU). If this approach were used, you could explain the lack of power to test for interactions across 5 time points. Because the pandemic interrupted the study, I think you also have a justifiable rationale for not collecting data from more participants.

• The use of a per-protocol analysis instead of intention-to-treat is potentially an issue because you lost more people from the assistive group than the main comparator group, perturbing. If my other comments are correct about the statistical analysis, I recommend revising the statistical approach to an intention-to-treat model, probably with a missing-at-random approach.

Minor comments

Overall

• One issue throughout the text is the nomenclature used to designate the different experimental groups. Specifically, the word perturbing is used for one group, but both the assistive and perturbing group experience perturbations. I believe interpretation of the text would be improved if you used to nomenclature on ClinicalTrials.gov: “error reduction,” “error augmentation,” and “control.”

Abstract

• From the abstract, it wasn’t clear what the overall training was, just what the different interventions were. Please add detail so readers can gather the overall intervention just from the abstract.

Introduction

• Overall, the introduction provides a good rationale for the study.

• No clear hypothesis is presented in the final paragraph of the Introduction. Please revise.

Methods

• Make sure details match those on ClinicalTrials.gov.

• Page 7, Line 134: How was it determined how many 3-minute trials they performed?

• Page 8, Line 163-164: The application of external force in the study depended on the prediction of a mechanically appropriate step width (see reference 6, 12). Presumably, there is potential for error with this prediction. Can you provide any insight?

• Please report how many in each group had external force applied to narrow vs. widen step width. Although the interventions were designed to either assist in achieving a mechanically appropriate step width or exacerbate errors in step width, this was again using model predictions, which may err. At the same time, we know that the external forces either narrowed or widened the step. Hence, your results could have occurred because the experimental groups differed in whether the step was narrowed or widened and the variability of such.

• Can you provide rationale for why the last minute of each trial was in transparent mode? Normally a “catch” trial would capture the first few steps when the perturbation was removed. Instead, 1/3rd of the intervention time was not actually the intervention.

• So, if they missed a session, it wasn’t just pushed to the end to ensure they achieved 24 training sessions? Can you please explain why the 24 training sessions had to be performed within a certain timeframe without flexibility?

Tables & Figures

• Table 2: please redo to show the actual values (including baseline). This will aid the reader in looking for baseline differences and not misrepresent the results.

• Figure 1: The presentation of transparent, assistive, and perturbing keys within the figure makes it look like the groups do 3 minutes of each, then 3 more minutes. I recommend moving this to the legend or a small legend.

• Figure 2: how is medical exclusions different from exclusion criteria?

• Figure 3 is good. It does look there may be some baseline differences between groups, hence, as discussed the statistical approach needs to be adjusted, or the manuscript needs to refrain from making between group comparisons (of changes).

Reviewer #2: This is a well-written description of a small, three-arm trial. I have a few minor comments.

1. The randomization block size was not specified. Were blocks randomized? What strategies were implemented to mitigate selection or ascertainment bias?

2. Were the assumptions underlying the sample size (accrued from a pilot study of n=56) realized? This should be clearly stated in the Discussion, as either a strength or a weakness.

Reviewer #3: This parallel, double-blinded RCT looked at the effects of perturbation training, a control condition, and assistive training on step modulation in PwCS. The perturbation group demonstrated improvements in paretic step width modulation which were sustained through a 12 week follow-up, whereas no change was noted for the control or assistive group. Despite promising results, the authors do a great job of explaining that there was not always a difference between the perturbation and control group, therefore the changes noted may have been due to a reduction in step width modulation at baseline for the perturbation group. This study demonstrates sound research design with good reasoning and interpretation of results. I have only minor concerns that can be easily clarified by the authors.

Introduction:

- Please clarify the authors’ hypotheses regarding the ‘assisted’ and ‘minimal external forces’ conditions? It’s mentioned that authors hypothesized the perturbations would cause a change in paretic step width modulation, but nothing about how this intervention would compare to the other two.

Materials and Methods:

- For clarity, it would help to have something signaling that transparent mode was the concluding minute of every other trial either in Figure 1 or in the caption

- More explanation of Paretic pPD is required for reader clarity (lines 217-222)

Results:

- Were statistical analyses completed to determine potential significant differences in participant characteristics?

Discussion:

- Lines 463-469 speculate that perhaps PwCS learn early in the early post-stroke period to walk with a wider base gait, and that practicing with perturbations allows them to realize that a more narrow gait is possible. However, there is no report of baseline step width listed. Was it the case that these participants demonstrated a larger than average step width during the pre-intervention assessment? And did it reduce post-intervention or at follow-up? If this wasn’t calculated, please clarify.

- What are your thoughts as to why there was a change in pSW for the perturbation group and not the assistance or control group for this specific intervention, when other studies you’ve sited (lined 469-475) have shown that both exaggerating and reducing errors or resistance can improve asymmetry in PwCS?

6. PLOS authors have the option to publish the peer review history of their article (what does this mean?). If published, this will include your full peer review and any attached files.

Reviewer #1: **Yes: **Brice Cleland

Reviewer #2: No

Reviewer #3: No

---

## [Author Response · Author response to Decision Letter 0]

2 Sep 2024

All reviewer comments are addressed in the uploaded Response to Reviewers document.

---

## [Decision Letter · Decision Letter 1]

24 Sep 2024

Effect of mediolateral leg perturbations on walking balance in people with chronic stroke: a randomized controlled trial

PONE-D-24-04911R1

Dear Dr. Dean,

We’re pleased to inform you that your manuscript has been judged scientifically suitable for publication and will be formally accepted for publication once it meets all outstanding technical requirements.

Kind regards,

Ryan T. Roemmich

Academic Editor

PLOS ONE

Additional Editor Comments (optional):

Thank you for submitting your revised manuscript to PLOS ONE. The three reviewers appreciated the effort put forth in your revisions, and we are now happy to accept the paper for publication.

Reviewers' comments:

Reviewer's Responses to Questions

**Comments to the Author**

1. If the authors have adequately addressed your comments raised in a previous round of review and you feel that this manuscript is now acceptable for publication, you may indicate that here to bypass the “Comments to the Author” section, enter your conflict of interest statement in the “Confidential to Editor” section, and submit your "Accept" recommendation.

Reviewer #1: All comments have been addressed

Reviewer #2: All comments have been addressed

Reviewer #3: All comments have been addressed

2. Is the manuscript technically sound, and do the data support the conclusions?

Reviewer #1: Yes

Reviewer #2: (No Response)

Reviewer #3: Yes

3. Has the statistical analysis been performed appropriately and rigorously? 

Reviewer #1: Yes

Reviewer #2: (No Response)

Reviewer #3: I Don't Know

4. Have the authors made all data underlying the findings in their manuscript fully available?

Reviewer #1: Yes

Reviewer #2: (No Response)

Reviewer #3: Yes

5. Is the manuscript presented in an intelligible fashion and written in standard English?

Reviewer #1: Yes

Reviewer #2: (No Response)

Reviewer #3: Yes

6. Review Comments to the Author

Reviewer #1: The reviewers have appropriately considered all reviewer comments and made improvements to the manuscript.

Reviewer #2: (No Response)

Reviewer #3: (No Response)

7. PLOS authors have the option to publish the peer review history of their article (what does this mean?). If published, this will include your full peer review and any attached files.

Reviewer #1: **Yes: **Brice Cleland

Reviewer #2: No

Reviewer #3: No

---

## [Editor Report · Acceptance letter]

29 Sep 2024

PONE-D-24-04911R1 

PLOS ONE

Dear Dr. Dean, 

I'm pleased to inform you that your manuscript has been deemed suitable for publication in PLOS ONE. Congratulations! Your manuscript is now being handed over to our production team.

Kind regards, 

on behalf of

Dr. Ryan T. Roemmich 

Academic Editor

PLOS ONE